# A Multi-phase Biogeochemical Model for Mitigating Earthquake-Induced Liquefaction via Microbially Induced Desaturation and Calcium Carbonate Precipitation

Caitlyn A. Hall[1,2,5,7], Andre van Turnhout[3,4], Edward Kavazanjian, Jr.[5], Leon A. van Paassen[5,6,7], and Bruce Rittmann[5,7]

[1]The Honors College and Biosystems Engineering Department, The University of Arizona, Tucson, AZ, USA
[2] Biosystems Engineering Department, The University of Arizona, Tucson, AZ, USA
[3] VALCON, Utrecht, The Netherlands
[4]Delft University of Technology, Delft, The Netherlands
[5]Center for Bio-mediated and Bio-inspired Geotechnics, Arizona State University, Tempe, AZ, USA
[6]Royal Boskalis Westminster N.V., Papendrecht, The Netherlands, USA
[7]Biodesign Swette Center for Environmental Biotechnology, Arizona State University, Tempe, AZ, USA

*Correspondence to*: Caitlyn A. Hall (cahall@arizona.edu)

**Abstract.** A next-generation biogeochemical model was developed to explore the impact of the native water source on microbially induced desaturation and precipitation (MIDP) via denitrification. MIDP is a non-disruptive, nature-based ground improvement technique that offers the promise of cost-effective mitigation of earthquake-induced soil liquefaction under and adjacent to existing structures. MIDP leverages native soil bacteria to reduce the potential for liquefaction triggering in the short term through biogenic gas generation (treatment completed within hours to days) and over a longer term through calcium carbonate precipitation (treatment completed in weeks to months). This next-generation biogeochemical model expands earlier modeling to consider multi-phase speciation, bacterial competition, inhibition, and precipitation. The biogeochemical model was used to explore the impact of varying treatment recipes on MIDP products and by-products in a natural seawater environment. The case study presented herein demonstrates the importance of optimizing treatment recipes to minimize unwanted by-products (e.g., $H_2S$ production) or incomplete denitrification (e.g., nitrate and nitrite accumulation).

**Keywords:** biogeochemical modeling, liquefaction, denitrification, desaturation, precipitation

## 1 Introduction

Microbially induced desaturation and precipitation (MIDP) is a biogeotechnical technique that takes advantage of native subsurface denitrifying bacteria to mitigate earthquake-induced soil liquefaction (O'Donnell et al., 2017a, b; Pham et al., 2018). MIDP mitigates liquefaction in two ways: generation of nitrogen gas ($N_2$) and mineral precipitation (usually calcium carbonate, $CaCO_3$). The impact of MIDP on the soil system is illustrated in Figure 1. The generated $N_2$ desaturates the soil, increasing its compressibility and reducing the increase in pore water pressure during cyclic loading, which is the root cause of

earthquake-induced liquefaction. Carbonate precipitation increases the soil strength, thereby increasing the intensity of earthquake sharing necessary to trigger liquefaction. A primary benefit of MIDP for liquefaction mitigation is, being non-disruptive, it can be used underneath existing structures (O'Donnell et al., 2017a; Hall, 2021). Trillions of dollars of existing infrastructure is at risk due to the potential for liquefaction, and currently that risk cannot be mitigated in a cost-effective way. MIDP is currently being evaluated at different experimental scales as a solution to this problem (O'Donnell et al., 2017a, b;

Moug et al., 2022).

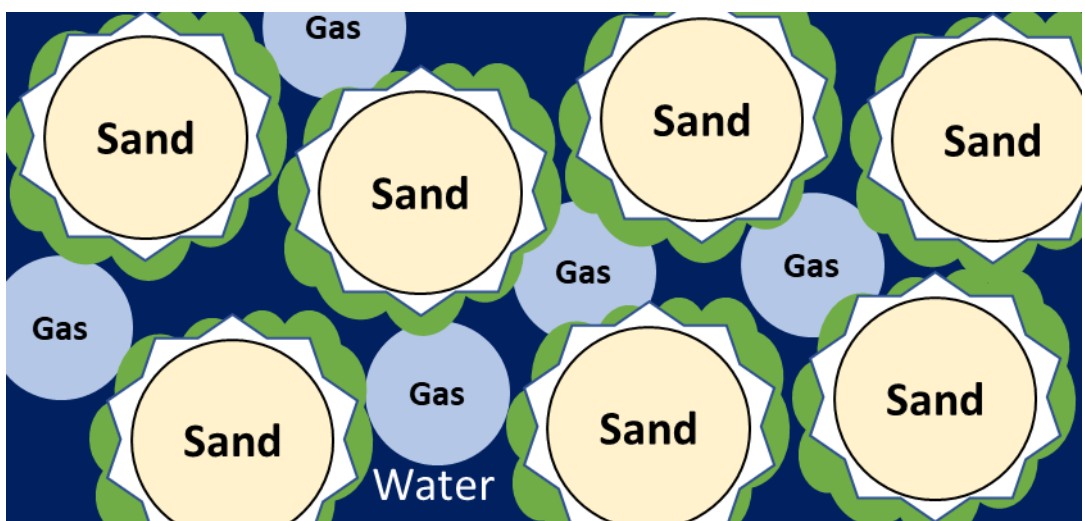

**Figure 1. Pore-scale illustration of MIDP in water (dark blue) saturated sand (light yellow), detailing the gas (light blue), calcium carbonate (white), and biomass (green) production.**

For environmental and economic reasons, we assume that local source water will be used to prepare the MIDP

treatment solution (i.e., dissolve the substrates needed to induce denitrification) in practice. Therefore, in developing this technique we must consider how naturally occurring constituents in the source water may affect denitrification. For example, a competing electron acceptor (e.g., sulfate) may consume the electron donor (e.g., acetate), leading to incomplete denitrification and the formation of unwanted products (e.g., hydrogen sulfide). In addition, the biogeochemical reactions during MIDP result in changes to alkalinity and pH that may alter process kinetics and final MIDP products.

To quantify the impact of source-water composition and to investigate how complex biogeochemical interactions occur during field application, we developed a next-generation biogeochemical model of MIDP. This mathematical model

expands upon previous modeling of MIDP, which did not consider the impact of source water on MIDP or the impact of MIDP on the aqueous subsurface environment (Pham, 2017; O'Donnell et al., 2019). Our next-generation MIDP model includes all essential biogeochemical processes based on the constituents commonly observed in the natural groundwater environments, substrates added to stimulate MIDP, and mechanisms that lead to desaturation and precipitation: e.g., $N_2$-gas formation, acid-base speciation, and $CaCO_3$ precipitation. Since MIDP often is deployed in coastal areas (due to the prevalence of liquefiable soil deposits in this environment), we include conditions typical for coastal seawater in our model.

## 2 Model Foundation

The next-generation model builds upon previous MIDP models (Pham, 2017; O'Donnell et al., 2019), but broadens the range of processes considered by the model. Our next-generation model considers microbial growth and decay, alternative microbial metabolic processes, gas production, mineral-solids production, alkalinity and pH, microbial inhibition, and desaturation and precipitation in both fresh water and coastal environments. A comparison of the components and processes considered by the two earlier MIDP models and our next-generation model is provided in Table A1 in Appendix A.

The next-generation model was constructed in Matlab (Little and Moler, 2017), and the code and necessary files are publicly available online at doi.org/10.5281/zenodo.7410676. The modeling equations (e.g., microbial growth, $CaCO_3$ precipitation, and biogenic gas evolution) were programmed within the original, publicly available van Turnhout Toolbox, a general-form mechanistic batch model for environmental systems that considers species in the gas, liquid, and solid phases (van Turnhout et al., 2016). The van Turnhout Toolbox is a program that includes a system of ordinary differential equations that model biogeochemical reactions. The van Turnhout Toolbox is coupled with ORCHESTRA, an extensive database of established geochemical equilibria based on MINTEQ, to simulate chemical speciation during said modeling (Meeussen, 2003). The MIDP-specific biogeochemical model components (i.e., stoichiometry, type of inhibition and kinetics, potential chemical species) were specified in an input spreadsheet that the program accesses. The degree of saturation and percent (by weight) of mineral precipitation were calculated outside of the van Turnhout Toolbox using model results, as discussed in Section 3.2 of this paper.

The Toolbox's logic flow and calculation sequence are as follows (Meeussen, 2003; van Turnhout et al., 2016), using $H_2CO_3$, $HCO_3^-$, $CO_3^{2-}$, $H^+$, and $OH^-$ to illustrate the process for the carbonate system.

1. At t = 0, the program loads the input concentrations file, which includes the concentration of all total species (e.g., $H_2CO_3$ representing DIC, $H^+$) and the stoichiometry for metabolic and kinetic reactions:  e.g.,

$$0.222NO_3^- + 0.125C_2H_3O_2^- + 0.146H^+ \rightarrow 0.202NO_2^- + 0.147H_2CO_3 + 0.103CH_{1.8}O_{0.5}N_{0.2} + 0.021\,H_2O$$

2. Ordinary differential equations are used to determine compound consumption and production based on the reaction stoichiometry and kinetic equations (e.g., precipitation, biotransformation, and mass transfer) at each time step. For

example, the graphic in Figure 2 illustrates that, as $C_2H_3O_2^-$ is consumed from microbial consumption, $H_2CO_3$ is produced.

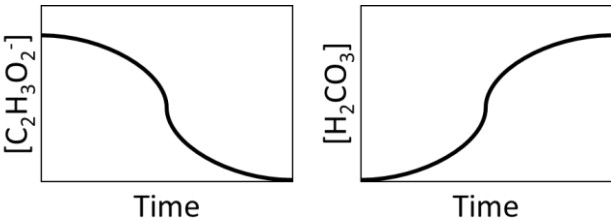

**Figure 2. Illustration of consumption of the consumption of $C_2H_3O_2^-$ and production of $H_2CO_3$ during MIDP.**

3. At each time step, the following set of linear equations is solved to determine the relative derived concentrations of $H_2CO_3$, $HCO_3^-$, $CO_3^{2-}$, $H^+$, and $OH^-$ from $H_2CO_3$ produced in the previous steps. This is done in the ORCHESTRA biochemical module.

   a. Mass balance equations – the left side of the equation is the total dissolved inorganic carbon, $H_2CO_3$, from

the stoichiometry described in steps 1 and 2. The right side are the derived concentrations of species as a result of speciation and indicated with italics.

$$H_2CO_3 = H_2CO_3 + HCO_3^- + CO_3^{2-}$$

   b. *Electroneutrality* – all potentially produced charged species related to this balance are considered.

$$H^+ = OH^- + HCO_3^- + CO_3^{2-}$$

c. Acid equilibrium for $H_2CO_3$

$$K_a = \frac{[CO_3^{2-}][2H^+]}{[H_2CO_3]}$$

   d. Acid equilibrium for $HCO_3^-$

$$K_a = \frac{[CO_3^{2-}][H^+]}{[HCO_3^-]}$$

   e. Water equilibrium

$$K_w = [OH^-][H^+] = 1.0 \cdot 10^{-14}$$

4. pH is calculated based on the derived $H^+$ concentration based on solving simultaneously 3(a-e).

While the carbonate system is used here as an example, this stepwise process is used for all acid-base species and considers the total system set of reactions and species to achieve equilibrium. The total system electroneutrality considered in the model for all considered species is as follows:

$$H^+ + NH_4^+ + Ca^{2+} + CaHCO_3^+ + CaOH^+ + CaC_2H_3O_2^+ + Fe^{3+} + Fe^{2+} + FeOH^+ + Fe(OH)_2^+ + Na^+$$
$$= OH^- + HCO_3^- + CO_3^{2-} + NO_3^- + NO_2^- + C_2H_3O_2^- + SO_4^{2-} + HSO_4^- + HS^- + Cl^-$$

The compounds listed above are defined in the input spreadsheet as possible species. The model then uses ORCHESTRA to determine the concentration of each species based on the biogeochemical reactions and system equilibrium.

## 3 Model Principles

Denitrification is a multi-step process of nitrogen-species reduction. During each reduction step, energy and biomass are produced when paired with oxidation of an electron donor that produces a thermodynamically favorable reduction-oxidation (redox) reaction. The four steps of nitrogen reduction in denitrification conform to the following reduction half reactions, each consuming two or one electron equivalent ($e^-$) (Rittmann and McCarty, 2020):

Nitrate Reduction to Nitrite
$$0.5NO_3^- + 2H^+ + 2e^- \rightarrow 0.5NO_2^- + 0.5H_2O$$

Nitrite Reduction to Nitric Oxide
$$NO_2^- + 2H^+ + e^- \rightarrow NO + H_2O$$

Nitric Oxide Reduction to Nitrous Oxide
$$NO + H^+ + e^- \rightarrow 0.5N_2O + 0.5H_2O$$

Nitrous Oxide Reduction to Dinitrogen
$$0.5N_2O + H^+ + e^- \rightarrow 0.5N_2 + 0.5H_2O$$

In the model developed herein, the four steps were simplified to two steps, nitrate to nitrite and nitrite to dinitrogen gas. The reductions of nitric oxide and nitrous oxide are assumed to occur completely, as they are thermodynamically favorable (Appendix A). In contrast, the accumulation of $NO_2^-$ must be considered explicitly because it is a denitrification inhibition 'bottleneck' and presents a risk to human health (Pham et al., 2018; Almeida et al., 1995). The variables for the equations used in the model described in the next several sections are detailed in Table 1.

**Table 1. Notation, units, and description for variables used in the biogeochemical model**

| Symbol | Units | Description |
|---|---|---|
| $\hat{q}$ | $mol_{donor}\ mol_{biomass}^{-1}\ d^{-1}$ | maximum specific rate of electron-donor utilization |
| $X_a$ | $mol\ L^{-1}$ | active biomass concentration |
| $C_d$ | $mol\ L^{-1}$ | electron donor concentration |
| $K_d$ | $mol\ L^{-1}$ | electron donor half-maximum substrate concentration of the |
| $C_a$ | $mol\ L^{-1}$ | electron acceptor concentration |
| $K_a$ | $mol\ L^{-1}$ | electron acceptor half-maximum substrate concentration |
| $I_i$ | | inhibition factor ($0 < I_i < 1$) |
| $Y$ | $mol_{biomass}\ mol_{donor}^{-1}$ | biomass yield from consumed electron donor substrate |
| $\mu_{max}$ | $d^{-1}$ | maximum specific microbial growth rate; product of $Y$ and $\hat{q}$ |
| $b$ | $d^{-1}$ | endogenous decay |
| $K_i$ | $mol\ L^{-1}$ | inhibition constant |
| $C_i$ | $mol\ L^{-1}$ | concentration of the inhibiting species |
| $v_{i[g]}$ | $mol\ L^{-1}\ d^{-1}$ | transfer rate from the aqueous phase to the gas phase |
| $k_L a$ | $d^{-1}$ | mass transfer rate constant |
| $Ci[g]$ | $mol\ L^{-1}$ | gas phase concentration of the gas species $i$ |
| $Ci[aq]$ | $mol\ L^{-1}$ | aqueous phase concentration of the biogenic gas species $i$ |
| $K_H$ | $L\ atm\ mol^{-1}$ | Henry's Law constant |
| $R$ | $L\ atm\ mol^{-1}\ K^{-1}$ | universal gas constant |
| $T$ | $K$ | system's absolute temperature |
| $[N_2]_g$ | $mol\ L_{pore}^{-1}$ | produced $N_2$ gas during MIDP |
| $[CO_2]_g$ | $mol\ L_{pore}^{-1}$ | produced $CO_2$ gas |
| $p$ | $atm$ | pressure at treatment depth |
| $S_g$ | $L_{gas}\ L_{pore}^{-1}$ | gas saturation level |
| $l$ | $L_{aq}\ L_{pore}^{-1}$ | aqueous solution in the pore space |
| $p_{N_2}$ | $atm$ | partial pressure of $N_2$ gas |
| $K_{H,N2}$ | $L_{aq}\ atm_{N2}\ mol_{N2}^{-1}$ | Henry's constant for $N_2$ at standard temperature |
| $p_{CO_2}$ | $atm$ | partial pressure of $CO_2$ gas |
| $K_{H,CO2}$ | $L_{aq}\ atm_{CO2}\ mol_{N2}^{-1}$ | Henry's constant for $CO_2$ at standard temperature |
| $Y_{NO3^-}$ | $mol_{NO3^-}\ mol_{donor}^{-1}$ | stoichiometric coefficients of $NO_3^-$ |
| $Y_{N2}$ | $mol_{N2}\ mol_{donor}^{-1}$ | stoichiometric coefficients of $N_2$ |
| $Y_{CO2}$ | $Mol_{CO2}\ mol_{donor}^{-1}$ | stoichiometric coefficients of $CO_2$ |
| $\varphi$ | $L_{pore}\ L_{total}^{-1}$ | soil porosity |
| $R_p$ | $mol\ L^{-1}\ d^{-1}$ | net rate of precipitation ($R_p > 0$) or dissolution ($R_p < 0$) of minerals |
| $ka$ | $d^{-1}$ | combined coefficient for constant mineral growth rate and the average crystal surface area |
| $K_{sp}$ | $mol^2\ L^{-2}$ | constant solubility product. |
| $Y_{CaCO3}$ | $mol\ CaCO_3\ mol_{donor}^{-1}$ | $CaCO_3$ yield |
| $NO_3^-{}_d$ | $mol_{NO3}\ L_{pore}^{-1}$ | $NO_3^-$ needed to achieve the target desaturation |
| $[NO_3^-]_C$ | $mol\ L_{pore}^{-1}$ | $NO_3^-$ needed to achieve the target $CaCO_3$ |
| $e$ | $L_{pore}\ L_{soil}^{-1}$ | void ratio |
| $\rho_{soil}$ | $kN\ L_{soil}^{-1}$ | soil density |
| $u_{CaCO3}$ | $g\ CaCO_3\ mol^{-1}\ CaCO_3$ | molarity to molecular weight conversion coefficient |

### 3.1 Microbial Metabolism, Growth, and Decay

The processes within the model follow Monod kinetics, represented as multiplicative dual-substrate limitation (O'Donnell et al., 2019; Bae and Rittmann, 1996).

$$\frac{dC_d}{dt} = -\hat{q}X_a \frac{C_d}{K_d+C_d} \cdot \frac{C_a}{K_a+C_a} I_i \tag{1}$$

Eq. 1 considers the electron-donor substrate ($C_d$) and three electron-acceptor substrates ($C_a$): $NO_3^-$ and $NO_2^-$ for denitrifying bacteria and $SO_4^{2-}$ for sulfate-reducing bacteria. For preliminary analysis, we assumed an initial denitrifier biomass concentration of 0.5 mmol $L^{-1}$ and sulfate-reducing biomass concentration of 0.25 mmol $L^{-1}$. The values of the constants ($\hat{q}$, $K_d$, and $K_a$) are in Appendix A, along with the derivations of said parameters based on Rittmann and McCarty (2020). Appendix A also includes preliminary model results assuming a reduced initial denitrifier biomass concentration of 0.05 mmol $L^{-1}$ and sulfate-reducing biomass concentration of 0.025 mmol $L^{-1}$. The comparison of the findings detailed in the results section of this paper and those found in Appendix A illustrate the influence of the initial biomass concentration on the model results. The main impact of lowering the initial biomass concentration was delayed start-up time for the microbiological processes, though the trends remained the same for all modeled scenarios. These derivations were used to determine reaction stoichiometry, true yield ($Y$) and $\mu_{max}$ (maximum specific growth rate) for all electron-donor and -acceptor pairs and the nitrogen source. The values of the kinetic and stoichiometric parameters are detailed in Table 2. The inhibition factor $I_i$ is described in a later section of this paper.

Values of half-maximum-rate concentrations ($K_d$ and $K_a$) in the literature show variability for each electron donor and acceptor pair due to the wide range of environments of the microorganisms (e.g., sediment, estuarine water, wastewater) and the high degree of diversity of microorganisms able to carry out these reactions (Abdul-Talib et al., 2002; Papaspyrou et al., 2014; Vavilin and Rytov, 2015). Table 3 details the constants we used as representative values for each $K_d$ and $K_a$ (for Eq. 1) based on relevant electron-donor and -acceptor pairs and sources of those values. While these values are not specific to a coastal seawater environment, they have been experimentally validated.

**Table 2. Reaction stoichiometry, yield (*Y*), and maximum specific growth rates ($\mu_{max}$) expected during MIDP, considering acetate as the electron donor and natural electron acceptors. Units for all parameters are in Table 1.**

| Electron Acceptor | Nitrogen Source | $\hat{q}$ | Y | $\mu_{max}$ | Reaction Stoichiometry |
|---|---|---|---|---|---|
| Nitrate | Nitrate | 8.12 | 0.82 | 6.68 | $0.222NO_3^- + 0.125C_2H_3O_2^- + 0.146H^+$ $\rightarrow 0.202NO_2^- + 0.147H_2CO_3$ $+ 0.103CH_{1.8}O_{0.5}N_{0.2} + 0.021\,H_2O$ |
| Nitrite | Nitrate | 11.69 | 0.99 | 11.6 | $0.054NO_3^- + 0.202NO_2^- + 0.270C_2H_3O_2^- + 0.525H^+$ $\rightarrow 0.101N_2 + 0.272H_2CO_3$ $+ 0.268CH_{1.8}O_{0.5}N_{0.2} + 0.154H_2O$ |
| Sulfate | Nitrate | 3.74 | 0.58 | 2.18 | $0.015NO_3^- + 0.072SO_4^- + 0.125C_2H_3O_2^- + 0.284H^+$ $\rightarrow 0.072H_2S + 0.177H_2CO_3$ $+ 0.073CH_{1.8}O_{0.5}N_{0.2} + 0.015H_2O$ |
| Nitrate | Ammonium | 6.95 | 1.01 | 6.99 | $0.236NO_3^- + 0.125C_2H_3O_2^- + 0.025NH_4^+ + 0.10H^+$ $\rightarrow 0.236NO_2^- + 0.124H_2CO_3$ $+ 0.126CH_{1.8}O_{0.5}N_{0.2} + 0.050\,H_2O$ |
| Nitrite | Ammonium | 9.65 | 1.26 | 12.2 | $0.235NO_2^- + 0.261C_2H_3O_2^- + 0.066NH_4^+ + 0.431H^+$ $\rightarrow 0.118N_2 + 0.193H_2CO_3$ $+ 0.328CH_{1.8}O_{0.5}N_{0.2} + 0.249H_2O$ |
| Sulfate | Ammonium | 3.63 | 0.18 | 0.66 | $0.113SO_4^- + 0.125C_2H_3O_2^- + 0.005NH_4^+ + 0.346H^+$ $\rightarrow 0.113H_2S + 0.227H_2CO_3$ $+ 0.023CH_{1.8}O_{0.5}N_{0.2} + 0.009H_2O$ |

**Table 3. Half-maximum-rate concentrations, $K_d$ and $K_a$, used for each electron-donor and -acceptor pair**

| Electron Donor | $K_d$ | Reference | Electron Acceptor | $K_a$ | Reference |
|---|---|---|---|---|---|
| Acetate ($C_2H_3O_2^-$) | $1.0 \cdot 10^{-5}$ | (Jia et al., 2020) | Nitrate ($NO_3^-$) | $5.4 \cdot 10^{-5}$ | (Abdul-Talib et al., 2002) |
| Acetate ($C_2H_3O_2^-$) | $1.0 \cdot 10^{-5}$ | (Jia et al., 2020) | Nitrite ($NO_2^-$) | $2.4 \cdot 10^{-5}$ | (Abdul-Talib et al., 2002) |
| Acetate ($C_2H_3O_2^-$) | $7.1 \cdot 10^{-5}$ | (Ingvorsen et al., 1984) | Sulfate ($SO_4^-$) | $2.00 \cdot 10^{-4}$ | (Ingvorsen et al., 1984) |

Microbial growth within the model is represented via reaction kinetics and stoichiometry expressed in Eq. 2:

$$\frac{dX_a}{dt} = X_a Y \hat{q} - b \tag{2}$$

Biomass yields ($Y$) are listed in Table 2. For sulfate-reducing bacteria, $b$ was set to 0.03 d$^{-1}$, whereas it was set to 0.05 d$^{-1}$ for denitrifiers (Rittmann and McCarty, 2020). As a result of decay, $NH_4^+$ is released and can serve as a nitrogen source for denitrification. Since $NH_4^+$ is thermodynamically favorable over $NO_3^-$ as a nitrogen source, it is used first before $NO_3^-$ during denitrification using a user-defined switch. We used the inhibition function described in Section 3.2 as the switch to interrupt
biomass from using $NO_3^-$ as the nitrogen source in the presence of $NH_4^+$. Decay involves endogenous respiration, and we assumed that 80% of decayed biomass is available as an acetate for metabolism, while 20% becomes inert biomass (Rittmann and McCarty, 2020). The stoichiometry for decay is:

$$0.238CH_{1.8}O_{0.5}N_{0.2} + 0.012H_2CO_3 + 0.095H_2O \rightarrow 0.125C_2H_3O_2^- + 0.048NH_4^+ + 0.077H^+$$

## 3.2 Inhibition

The van Turnhout Toolbox has the capability to model different inhibition mechanisms, but we only used non-competitive inhibition during denitrification because the enzymes that perform nitrate, nitrite, and sulfate reduction are different and not self-inhibitory (Glass and Silverstein, 1998). Denitrification inhibition, which slows nitrate and nitrite reduction rates (Glass et al., 1997), was included for the reduction of nitrate to nitrite and nitrite to $N_2$ gas. $I_i$ is a general term for inhibition of either step, with $i$ indicating which reaction. The form of $I_i$, shown in Eq. 3 is for non-competitive inhibition, and the inhibition
coefficients for each inhibitor are found in Table 4:

$$I_i = \frac{K_i}{K_i + c_i} \tag{3}$$

**Table 4. Non-competitive inhibition coefficients ($K_i$)**

| Inhibiting Compound | Reduction Process Inhibited | $K_I$ (mol L$^{-1}$) | Source |
|---|---|---|---|
| HNO$_2$ | Nitrate | $2 \cdot 10^{-6}$ | (Ma et al., 2010) |
| HNO$_2$ | Nitrite | $8 \cdot 10^{-8}$ | (Glass et al., 1997) |
| Salinity (as NaCl) | Nitrate, nitrite | 0.51[a]; 0.78[b] | [a](Panswad and Anan, 1999) [b](Mariangel et al., 2008) |
| H$_2$S | Nitrate, nitrite | $6 \cdot 10^{-5}$ | (Pan et al., 2019) |
| NO$_3^-$ | Sulfate | $1 \cdot 10^{-3}$ | (Veshareh et al., 2021) |
| NO$_2^-$ | Sulfate | $1 \cdot 10^{-3}$ | (Veshareh et al., 2021) |

[a]Unacclimated environments were DI and drinking water, [b]Acclimated environments were groundwater and sea water

Although several inhibitors could affect MIDP, HNO$_2$ is the most important inhibitor of the MIDP process (Lilja and
Johnson, 2016). Significant inhibition (95% rate reduction)of overall denitrification has been reported at 0.04 mg HNO$_2$ L$^{-1}$, which primarily impacted the intermediate $NO_2^-$ reduction step (Glass et al., 1997; Abeling and Seyfried, 1992), and a 60%

decrease in $NO_3^-$ reduction at 0.08 mg $HNO_2$ $L^{-1}$ also was reported (Ma et al., 2010). Within the model, $HNO_2$ inhibits $NO_3^-$ and $NO_2^-$ reduction using different inhibition coefficients (Table 4). The inhibition by $HNO_2$ is driven by pH speciation because $NO_2^-$ is dominant at a pH of 3.4 and higher and $HNO_2$ is negligible for pH $\geq$ 7.6. However, only a small concentration of $HNO_2$ can have a significant impact on denitrification, which underscores the importance of pH and the accumulation of the intermediate $NO_2^-$.

Inhibition between nitrate and nitrite reductions has been identified, with the presence of nitrate having a larger effect on nitrite reduction than nitrite on nitrate reduction (Lilja and Johnson, 2016; Glass et al., 1997; Almeida et al., 1995; Soto et al., 2007). Nitrite accumulation increases in the presence of nitrate until nitrate is depleted, such that nitrite reduction becomes the dominant process (Glass and Silverstein, 1998). When only nitrite remains, the rate of nitrite reduction increases. However, others have described that, as long as the electrons are adequately provided by the electron donor, competitive inhibition between nitrate and nitrite reductions is not significant (Soto et al., 2007; Ma et al., 2010; van den Berg et al., 2017). Therefore, the model does not include competitive inhibition, although it naturally includes competition for the electron donor between nitrate and nitrite reductions through thermodynamic favorability of nitrate reduction over nitrite reduction.

The model applies different non-competitive inhibition constants for salinity (as NaCl) for nitrate and nitrite reduction because nitrite reduction is more sensitive to salinity than nitrate. Because the magnitude of inhibition depends on experimental conditions and adaptation of the microorganisms, the value of $K_i$ may differ for local conditions (Krishna Rao and Gnanam, 1990).

Hydrogen sulfide ($H_2S$) also can be inhibitory to denitrification (Pan et al., 2019). Nitrate, nitrite, and $N_2O$ reductions have been inhibited by $H_2S$, though the extent and sensitivity of reduction in the presence of $H_2S$ was experiment-dependent (Senga et al., 2006; Pan et al., 2013; Tugtas and Pavlostathis, 2007; Liang et al., 2020; Cardoso et al., 2006). Within the model, one aqueous-phase $H_2S$-inhibition constant was used for both $NO_3^-$ and $NO_2^-$ reduction steps.

A pH < 6 can significantly slow denitrification (Glass and Silverstein, 1998) by inhibiting enzyme activity (Šimek and Cooper, 2002) and microbial growth (Estuardo et al., 2008). When the pH goes higher than 8, enzyme activity also can be impeded, leading to reduced denitrification rates or incomplete denitrification. Incidents of a high pH often are temporary, as $CaCO_3$ precipitation in MIDP buffers the pH (Salek et al., 2015). The benefit of including a pH-inhibition function when predicting denitrification has been demonstrated, but the values of their governing parameters are environment-specific and require fitting (Estuardo et al., 2008). Within the model, we considered the indirect net effect of pH only through $HNO_2$ inhibition, which does not require environment-specific parameters because the concentration of $HNO_2$ is automatically calculated within the model structure.

### 3.3 Biogenic Gas Production

O'Donnell et al. (2019) considered the production of $N_2$ and $CO_2$ during denitrification but did not consider the varying subsurface stresses that would influence phase transfer. The relative concentrations of the produced biogenic gas can affect

the distribution of gas at depth, since the gases have different solubilities, as well as different stoichiometries for electron-donor consumption.

Our next-generation MIDP model includes mass-transfer kinetics for transfers of $N_2$, $CO_2$, and $H_2S$ from the aqueous phase to the gas phase (or from the gas phase). We considered gas-phase transfer kinetics because assuming instantaneous gas phase transfer clearly would be an oversimplification, based on the review on mass transfer of biologically driven gas production completed by Kraakman et al. (2011). $N_2$, $CO_2$, and $H_2S$ concentrations were modeled in the aqueous and gas phases. The rate of transfer of a gaseous compound from the aqueous phase to (or from) the gas phase, $v_{i[g]}$, depends on the gas's degree of super-saturation and a mass-transfer-rate coefficient (Salek et al., 2015):

$$v_{i[g]} = k_l a\left(C_{i[g]} - \frac{C_{i[aq]}RT}{K_H}\right) \tag{4}$$

We assigned $k_La$ values for $N_2$, $CO_2$, and $H_2S$ of 1 d$^{-1}$ (Shin et al., 2002), though the values can vary widely based on porous medium conditions and temperature. We did not include pore-scale kinetics. The aqueous concentrations of $CO_2$ and $H_2S$ depend on the pH, as described below.

The biogenic gas volume needed to achieve a target level of desaturation ($S_g$) by $N_2$ ($[N_2]_g$) and $CO_2$ ($[CO_2]_g$) was determined by:

$$[N_2]_g + [CO_2]_g = \frac{pS_g}{RT} \tag{5}$$

in which $p$ was assumed to be equal to the sum of the hydraulic pressure at the treatment depth (7.6 m in an upcoming example) and the atmospheric pressure. Gas-phase $H_2S$ was not included in the desaturation calculations because its solubility is much higher than $N_2$ and $CO_2$.

Eq. 6 describes the amount of input $NO_3^-$ required for desaturation by $N_2$ and $CO_2$ ($NO_3^-{}_d$, mol$_{NO3}$ L$_{pore}^{-1}$) at the deepest target treatment depth, which is the lowest depth of the treated zone. The depth increases the pressure ($p_{N2}$ and $p_{CO2}$) and the needed amount of gas production to exceed the solubility threshold ($K_{H,N2}$ and $K_{H,CO2}$) and enter the gas phase, according to Henry's Law. The equation considers the amount of gas needed to overcome the solubility threshold to achieve the target level of desaturation (Hall et al., 2018; Pham, 2017):

$$NO_3^-{}_d = \frac{\left(\frac{[N_2]_g}{l} + \frac{p_{N_2}}{K_{H,N_2}}\right)Y_{NO_3^-}}{Y_{N_2}} + \frac{\left(\frac{[CO_2]_g}{l} + \frac{p_{CO_2}}{K_{H,CO_2}}\right)Y_{NO_3^-}}{Y_{CO_2}} \tag{6}$$

$K_{H,N2}$ was set to 1600 and $K_{H,CO2}$ to 29 (L atm mol$^{-1}$). We determine the amount of nitrate needed to achieve target desaturation levels at the lowest depth, because greater depth requires a higher concentration of gas to achieve target desaturation levels, as the pressures are at their respective maxima.

Eq. 7 was used to determine the biogenic gas volume ($V_g$, L$_{gas}$ L$_{tot}^{-1}$),

$$V_g = \frac{S_g RT \varphi}{p} \tag{7}$$

## 3.4 Solids Precipitation and Dissolution

Calcium carbonate precipitation occurs when dissolved inorganic carbon (DIC), produced from microbial substrate conversion

of the electron donor, exceeds the solubility of $CaCO_3$ for the concentration of $Ca^{2+}$ present. The stoichiometry for $CaCO_3$ precipitation is:

$$Ca^{2+} + H_2CO_3 \rightarrow CaCO_3 + 2H^+$$

At the beginning of each run, the concentration of species available for precipitation are calculated as their dissolved form as discussed in Section 2, then solids precipitation is determined. The van Turnhout Toolbox considers precipitation based on equilibrium calculations from the ORCHESTRA module (Meeussen, 2003). This assumption is valid when the rates of

250 precipitation and dissolution of minerals are much faster than the phase transfer between the aqueous and solid phases (Salek et al., 2015). Previous MIDP modeling did not consider precipitation kinetics, but assumed instantaneous equilibrium (Pham, 2017; O'Donnell et al., 2019). Instantaneous equilibrium may be an over-simplification for environmental conditions (Singurindy et al., 2004) in which the mechanisms of crystal nucleation, crystal growth, and mass transfer of reactants to the contact point of crystal growth are important (Rittmann et al., 2002). Therefore, we included precipitation and dissolution

kinetics in the next-generation model.

The model considers first-order precipitation and dissolution kinetics with respect to the $Ca^{2+}$ and $CO_3^{2-}$ concentrations (Rittmann et al., 2002; Chou et al., 1989):

$$R_p = ka \left(1 - \frac{K_{sp}}{[Ca^{2+}][CO_3^{2-}]}\right)[Ca^{2+}] \tag{8}$$

$K_{sp}$ was set to $1.83 \cdot 10^{-8}$ mol$^2$ L$^{-2}$ at 25°C for $CaCO_3$. $ka$ is a combined kinetic coefficient, because it is difficult to separate mass

transfer kinetics, crystal growth rate, and solid surface area (Rittmann et al., 2002; Spanos and Koutsoukos, 1998; Rittmann et al., 2003). $ka$ can have a large range depending on the environment and the ease of establishing precipitation nucleation points. We assumed $ka$ was 100 d$^{-1}$, though this value should be used as a fitting parameter subject to experimental validation. Precipitation was implemented using the van Turnhout Toolbox's method for biochemical reactions; $ka$ was specified as a reaction rate, and $K_{sp}$ was among the governing input parameters.

Eq. 9 was used to determine the amount of substrate needed to achieve a target precipitation level, which is determined by the ratio between mass of precipitated $CaCO_3$ and mass of the soil solids ($[CaCO_3]$, kg $CaCO_3$ kg soil$^{-1}$).

$$[NO_3^-]_c = \frac{[CaCO_3]\rho_{soil}Y_{NO_3^-}}{eu_{CaCO_3}Y_{CaCO_3}} \tag{9}$$

The stoichiometric coefficients considered the total amount of input $NO_3^-$ and produced $H_2CO_3$ for the total assumed two-step denitrification process. The DIC available for precipitation to provide $Y_{CaCO3}$ is estimated based on pH-driven speciation at each time step.

While we only considered calcium carbonate precipitation, the model has the flexibility to model precipitation of other minerals. The user would need to add in separate equations to model precipitation kinetics based on the reactants, desired products, and the $ka$ and $K_{sp}$ values appropriate for the desired precipitation reaction and product.

**3.5 Determining pH**

Because pH governs the concentration of important aqueous species based on acid/base speciation, the pH influences many of the geochemical reactions involved in MIDP. The pH was determined using the geochemical equilibrium software ORCHESTRA, which is part of the van Turnhout Toolbox. ORCHESTRA uses a mass balance on all species within the system and the products of rate-dependent processes as a function of time (i.e., kinetic, biogeochemical, and phase transfer processes). At each time step, the program performs a mass balance on all complexed species and their fate (e.g., transformed through microbial processes, precipitation, gas phase transfer) (van Turnhout et al., 2016; Meeussen, 2003).

**4 Case Study MIDP Behavior Seawater Conditions: Model Results and Discussion**

To demonstrate the capabilities of the model, we illustrate MIDP behavior when targeting desaturation for liquefaction mitigation in a coastal geochemical environment. In this case study, we demonstrate the impact of precipitation on biochemical reactions and the resulting multi-phase products and by-products resulting from MIDP and other environmental biogeochemical processes (e.g., sulfate reduction). However, we only consider desaturation as a target treatment mechanism and do not model an MIDP treatment recipe optimized for precipitation as a liquefaction-mitigation mechanism.

Table 5 details the chemical characteristics used to simulate coastal groundwater conditions, which were assumed to have the same characteristic of seawater due to intrusion (hereafter, referred to as "seawater"). The treatment substrate was added to the baseline level of these components. Table 5 does not reflect the varying concentrations of calcium acetate and calcium nitrate used in the treatment recipes, which are described later. We based the target treatment zone's soil properties on a case study of microbial desaturation via denitrification in Portland, Oregon presented by Moug et al. (2022). The deepest target treatment depth was 7.6 m. We assumed typical values of total unit weight, dry density, and porosity for uniform clean sand for the soil (Christopher et al., 2006): total unit weight of 19.5 kN m$^{-3}$ (dry unit weight of 15.6 kN m$^{-3}$; bulk density of 1950 kg m$^{-3}$) and porosity of 0.39.

**Table 5. Chemical Characteristics Assumed for a Coastal Seawater Environment**

| Compound | Coastal Seawater |
|----------|------------------|
| Nitrate | 20.3[1] μmol L$^{-1}$ |
| Nitrite | 0.14[1] μmol L$^{-1}$ |
| Sulfate | 28.2[2] mmol L$^{-1}$ |

| | |
|---|---|
| DIC | 2.13[1] mmol L$^{-1}$ |
| pH | 7.61[1] |
| Ammonium | 0.25[1] μmol L$^{-1}$ |
| Iron | 0.60[3] nmol L$^{-1}$ |
| Sodium | 0.47[2] mol L$^{-1}$ |
| Calcium | 10.3[2] mmol L$^{-1}$ |
| Chloride | 0.55[2] mol L$^{-1}$ |

[1]Average of measured values (Alin et al., 2017)

[2]Reference composition of "standard seawater" from and calculated for pH = 7.61 for acid-base species (Millero et al., 2008; European Commission. Directorate General for Research., 2011)

[3] (Bruland et al., 2001)

The reported desaturation levels required to increase the cyclic shear resistance for liquefaction mitigation range between 2 to 10% (He and Chu, 2014; O'Donnell et al., 2017a). We chose 10%, which is at the high end of the mitigation range, but well below the desaturation level at which gas starts to migrate upward or spread laterally, reported to occur at 20% for poorly graded (i.e., uniform) fine sands (Pham, 2017).

Following Eq. 5, 7.10 mmol L$^{-1}$ of total $N_2$ gas is required to meet a minimum target desaturation level of 10% throughout the entire treatment zone (assuming only desaturation via $N_2$ gas). To meet the target desaturation, we estimated the treatment recipe to be 22.4 mmol L$^{-1}$ of nitrate (1.84 g calcium nitrate L$^{-1}$) and 32.1 mmol L$^{-1}$ of acetate (2.54 g calcium acetate L$^{-1}$) using Eq. 6. However, based on background levels of nitrate and nitrite and the use of released ammonium as a nitrogen source, these levels were adjusted to establish the treatment recipe detailed in Table 6. The adjusted values were set 310 to result in complete denitrification (i.e., no residual nitrate or nitrite that relies on bacterial decay as the electron donor) and to not exceed 0.1 mmol L$^{-1}$ of acetate after complete denitrification. We compared the impact of varying the input levels of acetate (as calcium acetate) on the MIDP treatment to the matched treatment recipe. We tested the impact of addition of an extra 25% of acetate over our original estimations, referred to as the 'Excess Acetate' treatment recipe, and 25% less acetate from our original estimation, referred to as the 'Reduced Acetate' treatment recipe. For the excess- and reduced-acetate 315 comparisons, we did not adjust the input levels of nitrate (as calcium nitrate) from our original estimations of 22.4 mmol L$^{-1}$ (1.84 g calcium nitrate L$^{-1}$). We assumed that each treatment recipe was added in one application (i.e., not in a continuous flow-through manner).

**Table 6. MIDP treatment recipes for each modeled condition.**

| | Original Estimation | Matched | Excess Acetate | Reduced Acetate |
|---|---|---|---|---|
| Nitrate (mmol L$^{-1}$) | 22.4 (1.84 g calcium nitrate L$^{-1}$) | 19.9 (1.56 g calcium nitrate L$^{-1}$) | 22.4 (1.84 g calcium nitrate L$^{-1}$) | 22.4(1.84 g calcium nitrate L$^{-1}$) |
| Acetate (mmol L$^{-1}$) | 32.1 (1.84 g calcium nitrate L$^{-1}$) | 30.9 (1.77 g calcium acetate L$^{-1}$) | 40.1 (3.17 g calcium acetate L$^{-1}$) | 24.0 (1.90 g calcium acetate L$^{-1}$) |

The results of the matched treatment recipe on the subsurface gas volume and saturation profile are shown in the top two panels of

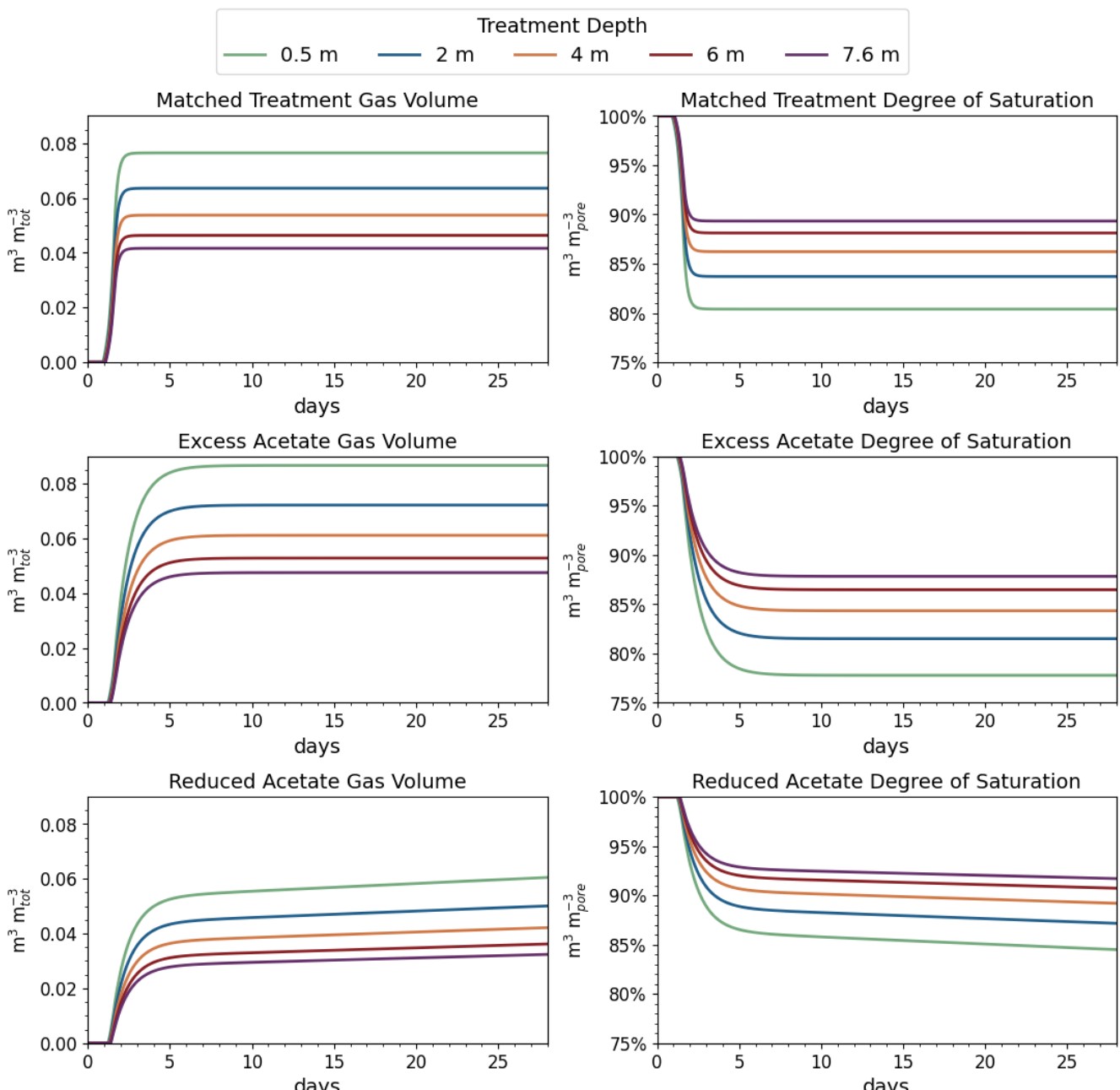

Figure 3. For the coastal seawater conditions, the target desaturation level of 10% at 7.6 m (or a degree of saturation of 90%)

was achieved by $N_2$ generation in approximately 2.0 days. The amount of $CO_2$ produced did not reach its saturation threshold, and $CO_2$ did not contribute to desaturation at any of the modeled depths. The difference in volume of gas at the different levels is due to the increase in hydrostatic pressure with depth.

The middle panels of Figure 3 indicate that adding excess acetate increased the degree of saturation at 7.6 m, which is shown by less than 90% saturation at 7.6 m. In contrast, the bottom two panels show that adding less acetate slowed $N_2$ generation so that 90% saturation was not reached at 7.6 m in 60 days.

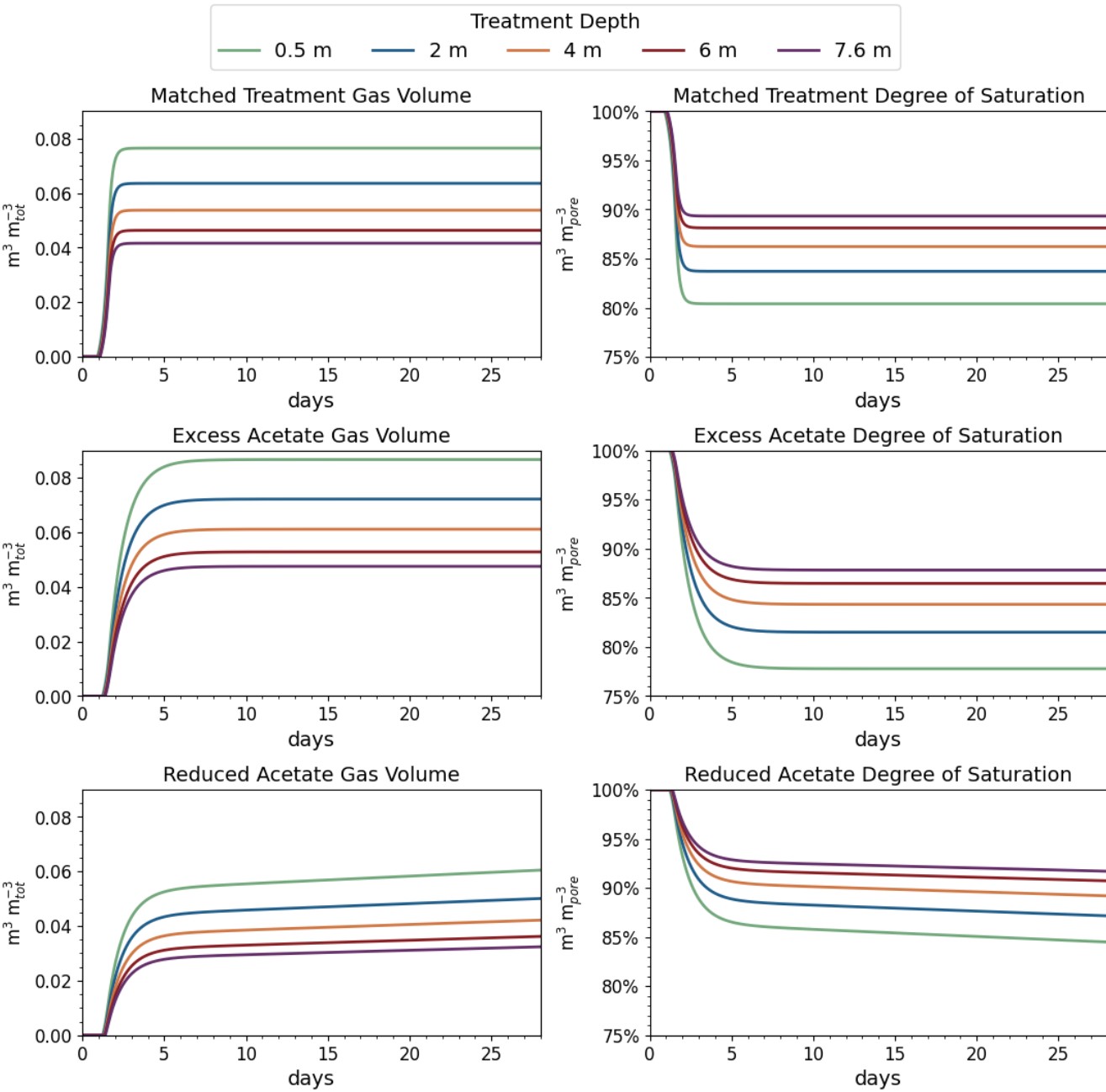

**Figure 3. Gas volumes normalized to the total soil volume (left) and degree of saturation by depth for the simulated Coastal seawater conditions. The desaturation target was 10%, or a saturation ratio of 90%.**

The impacts of the different MIDP treatment recipes on water quality are shown in Figure 4, and the initial five days of treatment for each modeled scenario are highlighted in Figure 5. With the matched-treatment recipe, almost all nitrate and nitrite were consumed by 1.6 days, and less than $10^{-4}$ mol L$^{-1}$ of acetate remained (Figure 5). Nitrite accumulation was transient and modest (3.3 mM at its peak); thus, complete denitrification was achieved with this treatment recipe for coastal seawater conditions (Figure 5). After 1.6 days, sulfate reduction began and continued to occur, driven by microbial endogenous respiration, at a small rate that resulted in the production of more total $CO_2$ and total $H_2S$ than the matched treatment over time (Figure 4).

With the excess-acetate recipe, all the nitrate was completely reduced, although small, transient accumulations of nitrite and nitrous acid occurred (Figure 5). As expected, not all the acetate was consumed with the excess-acetate recipe, and the remaining acetate led to sulfate reduction and the highest amount of produced $H_2S$ of the three modeled scenarios (Figure 4). Additional $N_2$ was produced because of the higher amounts of nitrate in the treatment recipe, leading to a level of desaturation at 7.6 m that exceeded the target 10% value.

For the reduced-acetate test, approximately 20% of the input nitrate remained after all the acetate was consumed, and this residual nitrate was slowly utilized beyond 1.5 days through biomass endogenous decay (Figure 4). The peak amount of accumulated nitrite was not as high as the other conditions because of the overall limited nitrate reduction, but some nitrite accumulation remained throughout the modeled 28 days due to the lack of acetate. The dip and quick increase in nitrous acid around 1.5 to 1.7 days (Figure 5) was due to the shift in electron donor from input acetate to bacterial decay. Not enough electron donor was available to reduce all of the input nitrate and the accumulated nitrite after 28 days (Figure 4), even though bacterial decay caused added denitrification. $N_2$ produced in the reduced-acetate condition did not meet the 10% desaturation threshold at 7.6 m at the end of the 28-day modeled period.

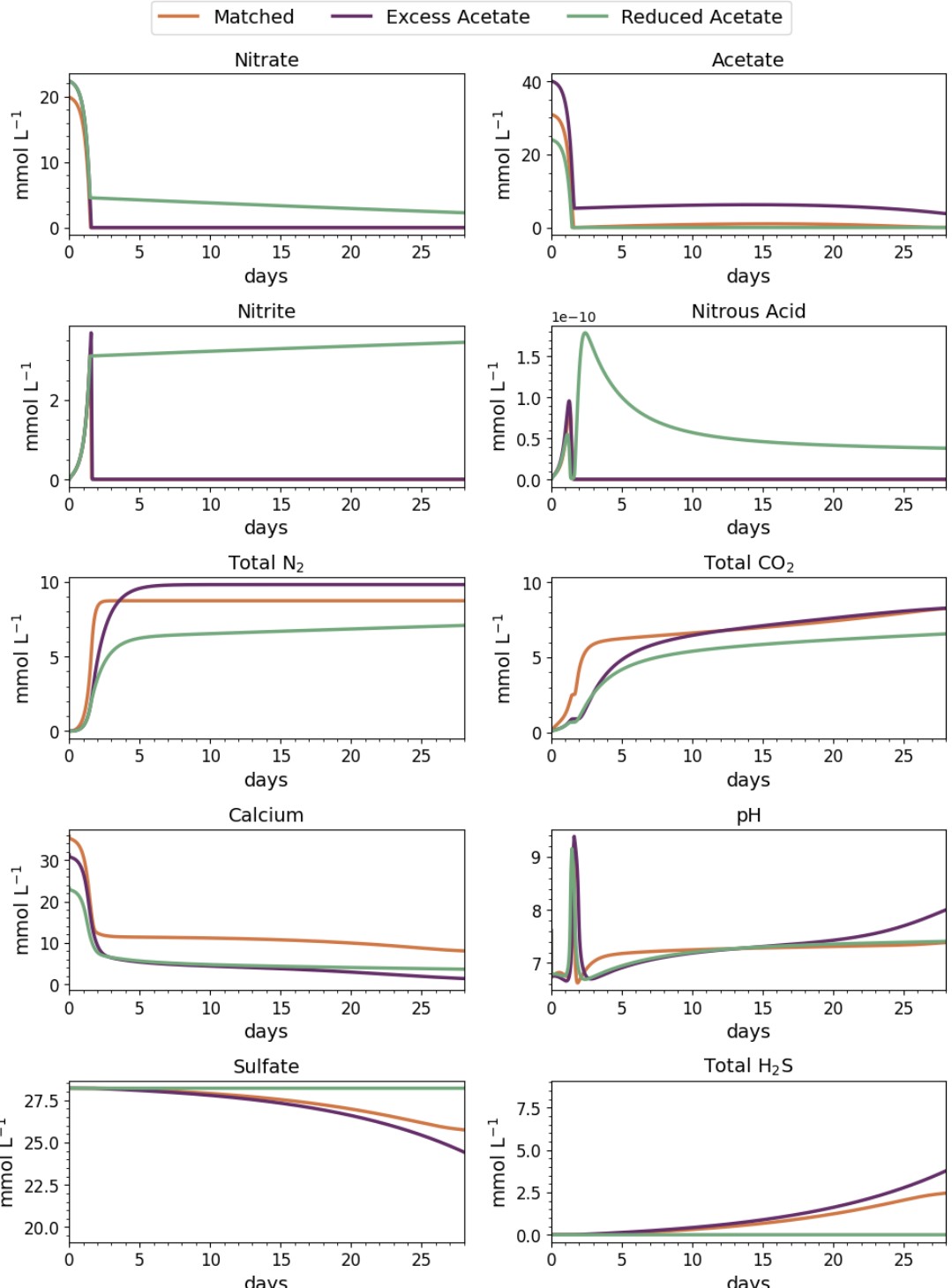

**Figure 4. Water-quality results for 28 days of MIDP in coastal seawater conditions targeting a desaturation level of 10% in three different treatment recipes: empirically matched, 25% excess acetate, and 25% reduced acetate.**

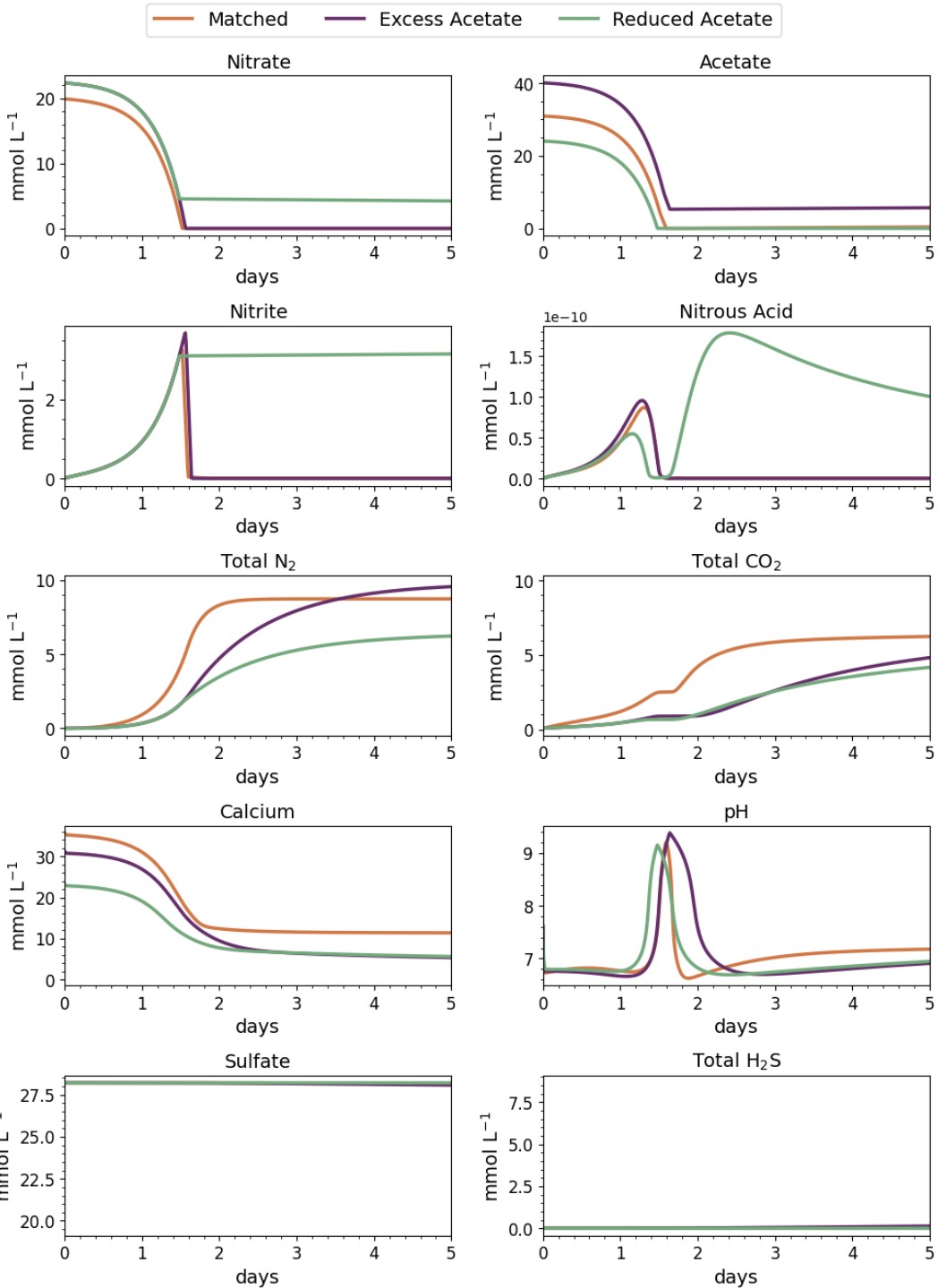

**Figure 5. Water-quality results for the first 3 days of MIDP in coastal seawater conditions targeting a desaturation level of 10% in three different treatment recipes: empirically matched, 25% excess acetate, and 25% reduced acetate.**

Nitrite reduction produced most of the base, which is demonstrated by the spike in pH after 1 and 1.4 days day in all scenarios (Figure 5), when the rate of nitrite reduction was at its maximum in each treatment. In the matched case, the pH returned to circumneutral after 1.8 days due to the precipitation of $CaCO_3$, which consumes base. This trend is reinforced by rapid $Ca^{2+}$ consumption in the early treatment time period (Figure 5), along with production of $CaCO_3$ in the first ~ 2 days, shown in the right panel of Figure 6. DIC production in the excess-acetate treatment lagged the matched recipe and was slightly

quicker in the reduced-acetate case, which also is seen with the pH trends in Figure 4 and Figure 5. However, the overall consumption of DIC and subsequent precipitation were slower in the excess- and reduced-acetate treatments, which correspond to the longer time for the pH to reach approximately neutral levels. The additional $CaCO_3$ precipitated with the excess acetate resulted from the excess of input calcium, since acetate was added as calcium acetate.

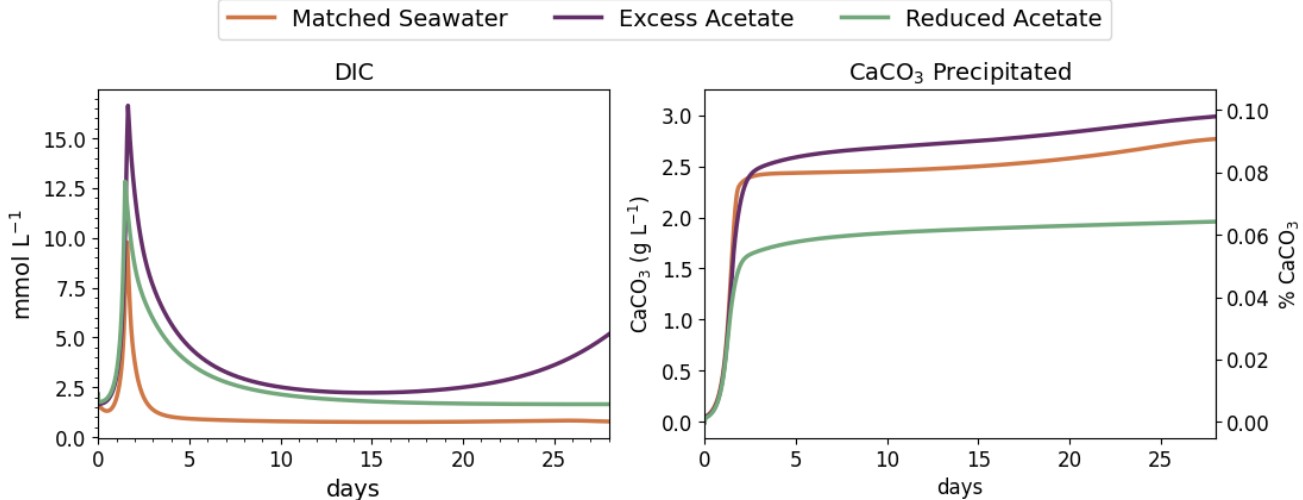

**Figure 6. DIC concentration and CaCO₃ precipitated during the first 3 days of MIDP in coastal seawater conditions targeting a desaturation level of 10%.**

        Microbial decay coupled to sulfate reduction also produced more DIC (left panel of Figure 6), promoting additional $CaCO_3$ precipitation after the completion of denitrification (after day 2 in Figure 6). In the excess-acetate and matched

treatments, an increase in DIC after 15 days was due to the increased rate of sulfate reduction because of the additional electron donor (Figure 4).

        In summary, the simulations show that the matched-acetate recipe optimized MIDP treatment targeted at desaturation for coastal seawater conditions by maximizing the desired outcome (i.e., $N_2$ production for desaturation) while minimizing undesired by-products (e.g., nitrite and nitrous-oxide accumulation from incomplete denitrification, residual acetate, and $H_2S$

from sulfate reduction).

## 5 Conclusion

The next-generation biogeochemical model expanded our previous biogeochemical models for MIDP by considering microbial stoichiometry and kinetics for two steps of denitrification and for sulfate reduction. The next-generation model also includes gas-liquid mass-transfer kinetics for $N_2$ and $CO_2$, $CaCO_3$ precipitation kinetics, microbial competition, and inhibition by $HNO_2$,

salinity, and sulfide. Model simulations demonstrated that adding nitrate and acetate using a properly matched recipe led to rapid desaturation without causing unwanted outcomes: incomplete desaturation and accumulations of nitrite and nitrous oxide with too-little acetate, or residual acetate and accelerated $H_2S$ generation with excess acetate. The model can be used to optimize treatment recipes for maximizing desaturation or precipitation in most subsurface groundwater environments for liquefaction mitigation. However, field data describing the environmental biogeochemical characteristics (e.g., pH,

background chemical concentrations) for the most optimized results is necessary to understand the potential biogeochemical reactions and processes that may impact MIDP, and subsequently, liquefaction mitigation.

**Acknowledgements**

The work presented in this paper was funded by the US National Science Foundation (NSF) Engineering Reserch Center (ERC) program under collaborative agreement ERC-1449501. Any opinions, findings and conclusions or recommendations

expressed in this paper are those of the authors and not of the NSF.

**Author Contribution**

CH and AvT co-developed the model. CH drafted the first version of this paper with input from AvT, LvP, EK, and BR.

**Code Availability**

The next-generation model was constructed in Matlab and the code and necessary files are publicly available online at

doi.org/10.5281/zenodo.7410676 (CC 4.0).

**Competing Interests**

The authors have no competing interests to declare.

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
