# Peer review of "A Multi-phase Biogeochemical Model for Mitigating Earthquake-Induced Liquefaction via Microbially Induced Desaturation and Calcium Carbonate Precipitation"

_EGUsphere, 2022_

## Author Comment (AC1)

**RC #1**

We thank the reviewer for their comments, as they have substantially improved the draft. Below we present the original question, followed by our response.

1. Line 54. I like Table A.1 that summarizes the capabilities of different models. There is a paper that just was published in Water Resources Research by some of these same authors (https://doi.org/10.1029/2022WR032907) - should this be included?
   a. **Thank you for bringing this to our attention, we now include this paper in Table A.1 of our revision.**
2. Consider adding a schematic 'cartoon' showing a porous medium representative elementary volume with liquid, gas, biomass phases, relevant biogeochemical processes, etc.
   a. **We will add such a diagram to illustrate the process.**
3. 50, Sec. 2, Model Foundation. I am not familiar with the van Turnhout Toolbox. This is used to solve the coupled system on nonlinear ordinary equations, coupled with the nonlinear algebraic equations for aqueous speciation. Can you add a few sentences to explain more about the numerical techniques used?
a. **We added the following text to describe the Toolbox in Section 2 of the manuscript:**

**The modeling equations (e.g., microbial growth, $CaCO_3$ precipitation, and biogenic gas evolution) were programmed within the original, publicly available van Turnhout Toolbox, a general-form mechanistic batch model for environmental systems that considers species in the gas, liquid, and solid phase (van Turnhout et al., 2016). The van Turnhout Toolbox is a system of ordinary differential equations coupled with ORCHESTRA to simulated chemical speciation (Meeussen, 2003), an extensive database of established geochemical equilibria. The MIDP-specific biogeochemical model components (i.e., stoichiometry, type of inhibition and kinetics, potential chemical species) were specified in an input spreadsheet that the program accesses. The degree of saturation and percent (by weight) of mineral precipitation were calculated outside of the van Turnhout Toolbox using model results, as discussed in Section 3.2 of this paper.**

**The Toolbox's logic flow and calculation sequence are as follows (Meeussen, 2003; van Turnhout et al., 2016), using $H_2CO_3$, $HCO_3^-$, $CO_3^{2-}$, $H^+$, and $OH^-$ to illustrate the process for the carbonate system.**

**1. At t = 0, the program loads the input concentrations file, which includes the concentration of all total species (e.g., $H_2CO_3$ representing DIC, $H^+$) and the stoichiometry for metabolic and kinetic reactions:  e.g.,**

**2. Ordinary differential equations are used to determine compound consumption and production based on the reaction stoichiometry and kinetic equations (e.g.,**

precipitation, biotransformation, and mass transfer) at each time step.  For example, the graphic in Figure  illustrates that, as $C_2H_3O_2^-$ is consumed from microbial consumption, $H_2CO_3$ is produced.

[Figure]

Figure 1. Illustration of consumption of the consumption of C2H3O2- and production of H2CO3 during MIDP.

3.  At each time step, the following set of linear equations are solved to determine the relative derived concentrations of $H_2CO_3$, $HCO_3^-$, $CO_3^{2-}$, $H^+$, and $OH^-$ from $H_2CO_3$ produced in the previous steps and already present. This is carried out in the ORCHESTRA biochemical module.

    a.   Mass balance equations – the left side of the equation is the total dissolved inorganic carbon, $H_2CO_3$, from the stoichiometry described in steps 1 and 2. The right side are the derived concentrations of species as a result of speciation and indicated with italics.

$$H_2CO_3 = H_2CO_3 + HCO_3^- + CO_3^{2-}$$

    b.  Electroneutrality – all potentially produced charged species related to this balance is considered.

$$H^+ = OH^- + HCO_3^- + CO_3^{2-}$$

    c.   Acid-base equilibrium for $H_2CO_3$

$$K_a = \frac{[CO_3^{2-}][2H^+]}{[H_2CO_3]}$$

    d.  Acid-base equilibrium for $HCO_3^-$

$$K_a = \frac{[CO_3^{2-}][H^+]}{[HCO_3^-]}$$

e. Water equilibrium

$$K_w = [OH^-][H^+] = 1.0 \cdot 10^{-14}$$

4. pH is then calculated based on the derived $H^+$ concentration based on 3(a-e).

While the carbonate system is used here as an example, this stepwise process is used for all acid-base species and considers the total system set of reactions and species to achieve equilibrium. The total system electroneutrality considered in the model for all considered species is as follows:

$$H^+ + NH_4^+ + Ca^{2+} + CaHCO_3^+ + CaOH^+ + CaC_2H_3O_2^+ + Fe^{3+} + Fe^{2+} + FeOH^+ + Fe(OH)_2^+ + Na^+$$
$$= OH^- + HCO_3^- + CO_3^{2-} + NO_3^- + NO_2^- + C_2H_3O_2^- + SO_4^{2-} + HSO_4^- + HS^- + Cl^-$$

These compounds are user defined in the input spreadsheet, but are used within the model by ORCHETRA using the Minteq4 chemical database to determine species complexation.

4. 81, eqn (1). I believe there are other mathematical forms to account for the impact of an inhibitory compound (e.g., Haldane Kinetics). Why is this form selected? Does this form only account for the 'inhibition' due to presence of a competing electron acceptor?
   a. **We included this type of inhibition because the inhibition is non-competitive inhibition. We did not include Haldane kinetics because the compounds are not self-inhibitive, as might be valid for aromatic hydrocarbons and chlorinated solvents. We added the following text to reflect this comment (state where):**
   b. **"The van Turnhout Toolbox has the capability to model different inhibition mechanisms, but we only used non-competitive inhibition during denitrification because the enzymes that perform nitrate, nitrite, and sulfate reduction are different and not self-inhibitory (Glass and Silverstein, 1998)."**
5. 84. Can you comment on the assumed initial conditions for biomass of denitrifying and sulfate-reducing microbes? Are these 'typical'? I would expect that the simulation results might be highly sensitive to these values.

a. **These values can vary greatly depending on the environment, and the time to achieve the target treatment woulddepend on the initial active biomass concentration (i.e., more biomass leads to faster treatment). We added additional results showcasing the sensitivity of the model results to initial biomass concentration. This is in Figure Z, which is shown here, along with accompanying text.**

6. 111, Table 3. The half maximum-rate constants are in units of mole/liter. Converting to mmole/L and looking at the conditions for the example simulation (Table 6) it appears that the $K_d$, $K_a$ values may be much smaller than the aqueous concentrations so that the Monod terms reduce to zero-order rate expressions (max possible rate). This is just an observation and may warrant some sensitivity study since the half-max rate constants are highly variable in the literature.

   a. **W**e**e believe that the largest impact on the process would be the initial biomass concentration, not the K values. Thus, we performed a sensitivity analysis looking at varying concentrations of initial biomass (see the preceding response).**

7. 119-120. Does the model formulation automatically switch between electron acceptors that are more thermodynamically favorable? How does the model switch to using ammonium as the electron acceptor?

   a. **We added additional text to address this question [where?]:**

   b. **"Since $NH_4^+$ is thermodynamically favorable over $NO_3^-$ as a nitrogen source, it is used before $NO_3^-$ during denitrification; this is implemented using a user-defined switch functiondescribed in Section 3.2; it stops biomass from using $NO_3^-$ as the nitrogen source in the presence of $NH_4^+$."**

8. 135. The text states that Ki value is the same for inhibition of nitrate and nitrite reduction by nitric acid, however Table 4 has different values. Please explain.

   a. **Thank you for catching this. It was a mistake that we corrected in Table 4.**

9. 3.3. I like the explanation in this section about computing the gas volume required to achieve target desaturation.

   a. **Thank you!**

10. 172. Sec. 3.3. As noted, the mass transfer coefficients are lumped values that are a function of the liquid-gas interfacial area. Therefore, I would expect there to be a dependence on the gas saturation. The sentence "We did not include pore-scale kinetics" is not clear. Does this sentence mean that you did not account for changing interfacial area? Given the complexity and uncertainty in modeling kinetic mass transfer, why not just use equilibrium partitioning? Is there field or laboratory evidence that kinetics are needed? I would expect the mass transfer coefficient would also be a highly sensitive parameter. The default value assumed (5 per day) is from a paper on sewer networks. I recommend checking the groundwater remediation literature (e.g., air sparging) for more representative values.

a. **First, we added a sentence in our manuscript with the following to justify our inclusion of kinetics:**

b. **"We considered gas-phase-transfer kinetics because assuming instantaneous gas phase transfer clearly would be an oversimplification, based on the review on mass transfer of biologically driven gas production completed by Kraakman et al. (2011)."**

c. **Second, following the recommendations of the reviewer, we adjusted our $k_La$ value to be 1 d$^{-1}$ based on bioremediation literature in soils.**

11. 185. Should the symbol [NO3]_d be added to Table 1?

    a. **Thank you for catching this omission; it was added to Table 1.**

12. 204, Eqn (8). I do not understand the statement that this rate expression is first-order with- respect-to calcium concentration, since the product [Ca] [CO3] is in the denominator. Calcite precipitation and dissolution has been studied extensively in the geology/geochemistry literature and I suggest adding a few key citations (e.g., Chou, L., R. M. Garrels, and R. Wollast (1989), Comparative study of the kinetics and mechanisms of dissolution of carbonate minerals, Chem. Geol., 78, 269–282.). As noted, calcite precipitation is a complex process and there are several calcium carbonate polymorphs of different stability.

a. **We adjusted the text to address that precipitation is driven by $CO_3^{2-}$ concentrations, and we added the recommended citation in Section 3.4: The model considers first-order precipitation and dissolution kinetics with respect to the $Ca^{2+}$ and $CO_3^{2-}$ concentrations (Rittmann et al., 2002; Chou et al., 1989)...**

13. 217, Sec. 3.5. Does the modeling framework allow for the presence of other mineral phases at equilibrium with the aqueous solution?

    a. **Yes! We added the following text to explain the models ability,**

    b. **"While we only considered calcium carbonate precipitation, the model has the flexibility to model other mineral precipitation. The user would need to add in separate equations to model precipitation kinetics based on the reactants, desired products, and the *ka* and $K_{sp}$ values appropriate for the desired precipitation reaction and product."**

14. 255-260, Table 6. I am a little confused by the treatment recipes. I was expecting to see numbers in Table 6 that were 25% greater and lesser than the matched case. L. 258 implies that the matched nitrate equals 22.4 mmol/L, but the table shows 19.0 mmol/L.

    a. **The 25% excess and reduced acetate are based on our original estimation, following Eq. 6 in the text. We added a column in Table 6 to reflect this to reduce confusion.**

**Table 6. MIDP treatment recipes for each modeled condition.**

| | Original Estimation | Matched | Excess Acetate | Reduced Acetate |
|---|---|---|---|---|
| Nitrate (mmol L⁻¹) | 22.4 (1.84 g calcium nitrate L⁻¹) | 19.0 (1.56 g calcium nitrate L⁻¹) | 22.4 (1.84 g calcium nitrate L⁻¹) | 22.4 (1.84 g calcium nitrate L⁻¹) |
| Acetate (mmol L⁻¹) | 32.1 (1.84 g calcium nitrate L⁻¹) | 22.4 (1.77 g calcium acetate L⁻¹) | 40.1 (3.17 g calcium acetate L⁻¹) | 24.0 (1.90 g calcium acetate L⁻¹) |

15. 225, Sec. 4, Case Study. Table 5. Are there any solid mineral phases present at the start of the simulation?  Should the initial fluid composition be in equilibrium with solid phases? This equilibrium is then perturbed by the input of the treatment fluid? Is there any possibility of iron minerals precipitating?

    a.  **Please see the response in RC1, Comment 3.  We also added the following to the precipitation section:**

    b.  **"At the beginning of each run, the concentration of species available for precipitation are calculated as their dissolved form as discussed in Section 2, then solids precipitation is determined."**

    c.  **Second, since this is a batch model, how the treatment solution is added is not considered.   This is an important topic, but beyond the scope of this manuscript.**

    d.  **Finally, iron precipitation is possible within the model but was not discussed due to the low concentration of iron in the background environment (0.60 nmol L⁻¹).**

16. Figs 1 & 2 simulation results. Please refer to my general comments regarding sensitivity analysis.

    a.  **Thank you for your comments on sensitivity; please refer to RC1, comment 6.**

---

## Author Comment (AC2)

**RC #2**

We thank the reviewer for their comments, as they have substantially improved the draft. Below we present the original question, followed by our response.

1. Title: To my mind, the title poorly reflects the content of this paper. My first question was "Desaturation and Precipitation of what??". Maybe desaturation and precipitation are well-understood terms in groundwater research, but I had to read the introduction to figure out the main research question, and even then, it was not fully explicit. Perhaps a more comprehensive title could be along the lines of "A multi-phase biogeochemical model for improving soil stability through microbially-induced water desaturation and _calcium_ carbonate precipitation", or similar. Line 20 in the abstract should make it clear that we are talking about N2 gas here, since methane was the gas that first sprung to mind for me.
    a. **We adjusted the title to be the following: "A Multi-phase Biogeochemical Model for Mitigating Earthquake-Induced Liquefaction via Microbially Induced Desaturation and Calcium Carbonate Precipitation"**

2. The introduction jumps straight into the topic with very little relevant background. Key facts are missing, such as: Why is liquefaction important? How much of a problem is it in today's society? What methods are currently used to tackle it? Is MDIP treatment a one-off exercise, or does it involve continuous application. Is the basic idea of desaturation that production of N2 gas reduces the partial pressure of H2O(g), thus favoring subsurface evaporation? And so on. Without such essential background, I got a little lost here trying to find the rationale for this study and its wider implications.
    a. **We augmented the intro paragraph to read: "Microbially induced desaturation and precipitation (MIDP) is a biogeotechnical technique that takes advantage of native subsurface denitrifying bacteria to mitigate earthquake-induced soil liquefaction (O'Donnell et al., 2017a, b; Pham et al., 2018). MIDP mitigates liquefaction in two ways: generation of nitrogen gas (N$_2$) and mineral precipitation (usually calcium carbonate, CaCO$_3$). The generated N$_2$ desaturates the soil, increasing its compressibility and reducing the increase in pore water pressure during cyclic loading, which is the root cause of earthquake-induced liquefaction. The carbonate precipitation increases the soil strength, thereby increasing the intensity of earthquake sharing necessary to trigger liquefaction. A primary benefit of MIDP for liquefaction mitigation is, being non-disruptive, it can be used underneath existing structures (O'Donnell et al., 2017a; Hall, 2021). Trillions of dollars of existing infrastructure are at risk due to the potential for liquefaction, and there is currently no cost-effective way to mitigate that risk. MIDP may be a**

**potential solution and is currently being evaluated at different experimental scales (O'Donnell et al., 2017a, b; Moug et al., 2022).**

    b.

3. The basic set-up of the model is not described and instead refers to earlier studies (which I do not have time to read). Please supply some more basic information. Is this a batch model, or a reaction transport model (1-D. 2-D ?). After reading the results, I presume this is a batch model. If not, what are the model dimensions, physical set-up (e.g. solid/liquid/gas fractions of pore space, grid-spacing etc), boundary and initial conditions? I presume that the scenario tested is for an anaerobic environment? How realistic is this assumption in actuality?

    a. **We added more text surrounding the model set-up; please see RC1, Comment 3.**

4. Why is a model that explicitly includes biomass growth and decay favorable over one where biomass is treated implicitly? Such a model comes at the burden of significant additional parameterization, with some parameters arguably poorly known such as microbial decay (see comment below). It would be instructive too see how sensitive the results are to the decay constant.

    a. **We opted to consider decay explicitly to avoid over- and under-estimations from responses from environmental conditions (i.e., increased decay with increase in temperature), observed in implicit models. As such, we are better able to look at sensitivity and simplify the model. While this may be an oversimplification, we will include this sensitivity analysis in our supplementary information along with our analysis on initial biomass concentration.**

5. The model includes nitrate reduction to nitrite, and nitrite reduction to dinitrogen. It seems as though a key process is missing here, namely, anammox (anaerobic ammonium oxidation by nitrate, producing N2). Ammonium is produced from the decaying microbial detritus, and anammox is widespread in anaerobic aquatic environments. Why was this not considered, and would its inclusion impact the treatment recipes? Some careful discussion is needed here.

    a. **Due to the high concentration of added acetate and the lack of ammonium and nitrite accumulation, anammox is not relevant. Anammox is an autotrophic (i.e., not heterotrophic) process in which ammonium is the electron donor and nitrite is the electron acceptor. In our situation, acetate is the (heterotrophic) electron donor and nitrate or nitrite is the electron acceptor, while ammonium is present in small concentrations due only to decay of heterotrophic biomass. Thus, none of the features that make anammox important are present in our situation. .**

6. It is not clear to me whether, in the real world, the treatments would be continuously applied. The model results shown seem to imply that once NO3 is exhausted, SO4 gets depleted and H2S accumulates. How does this relate back to a real-life scenario? Are we to expect that H2S gas will be released through the pore space at some point?

   a. **If desaturation is the desired outcome, a single treatment appears to be sufficient for liquefaction mitigation in granular soil (O'Donnell et al., 2017a, b). However, if the persistence of desaturation is uncertain and carbonate precipitation sufficient to mitigate liquefaction is also desired, several treatments may be required. Further, gas starts to migrate upward or spread laterally at a desaturation level of ~20% for poorly graded (i.e., uniform) fine sands (Pham, 2017). Its migration varies depending on the site's geology and stratigraphy (van Paassen et al., 2017). Since we never reach that high of desaturation for any of the gasses produced at the depths we model, we expect all gas to remain in the subsurface.**

7. The microbial growth parameters are derived on a quasi-first-principles basis in the supplement, which is nice to see. However, the derivation of the mortality rate constants is not part of this treatment. How were these values constrained, and how sensitive are model outputs to these values?

   a. **We used the decay parameter to reflect the aggregate mortality of the biomass. This number was assumed to be 0.05 d-1 based on reported values by Rittmann and McCarty (2020).**

8. Is there a typo in Table 4? Second column "sulfide". Sulfide cannot be reduced, or have I misunderstood this?

   a. **Thank you for catching the typo. This has been changed to be "sulfate"**

9. L208: Ka is given in units of L/d. Ca2+ is in mol/L (Table 1). This seems to conflict with the rate units of mol/L/d (Table 1). Should Ka in fact be in /d?

   a. **Thank you for catching this. Yes, $k_a$ should be d$^{-1}$. We adjusted this throughout the manuscript.**

10. L229: Typo: consider
    a. **This has been fixedl thank you for catching the typo.**

11. L229: Only the treatment optimized for desaturation is tested. Is there a good reason why precipitation as a liquefaction-mitigation mechanism is ignored? This seems at odds with the main thrust of the manuscript since, up to this point, the focus is on

both mechanisms (including the title of the paper!). If calcite precipitation were the desired treatment, how does the model deal with the ensuing reduction in pore space?

    a. We are preparing a separate paper on precipitation because our conclusion from that analysis is that achieving precipitation is technically inconsistent with desaturation (too much gas, e.g.), and we want to make that point strongly in its own paper.

    b. The model is a batch and we have clarified this further in the manuscript.

    c. Our recent work shows that desaturation can be expected to persist for decades in most geologic environments where this technique if feasable. See our response to RC2, Comment 6.

12. Table 5: Seawater Ca2+ is ~ 10mM, but the results (Fig. 2) show that Ca2+ at the start of the simulation is in excess of 20 mM. Please explain. Maybe this all becomes clear with a clearer description of the model initial conditions etc. Is Table 5 the initial condition?

    a. **The higher concentration of available calcium is from the input treatment recipe. Both the electron donor and electron acceptor are introduced as calcium acetate and calcium nitrate. We have added the following to reduce confusion:**

    b. **"Table 5 does not reflect the added treatment recipes of varying concentrations of calcium acetate and calcium nitrate, which are described later."**

13. Results plots: I don't know how to interpret these plots since it is not clear whether the data represent a one-off addition of a treatment or a continuous flow-through.

    a. **We added the following to the text:**

    b. **"We assumed that each treatment recipe was added in one application (i.e., not in a continuous flow-through manner)."**